# Criteria for PAVM Reintervention

**DOI:** 10.3390/jcm13206104

**Published:** 2024-10-13

**Authors:** Adam Fish, Elizabeth Knight, Katharine Henderson, Jeffrey Pollak, Todd Schlachter

**Affiliations:** Department of Interventional Radiology, Yale School of Medicine, New Haven, CT 06520, USAkatharine.henderson@yale.edu (K.H.); jeffrey.pollak@yale.edu (J.P.); todd.schlachter@yale.edu (T.S.)

**Keywords:** hereditary hemorrhagic telangiectasia, PAVM recurrence, PAVM paradoxical embolization, PAVM reintervention

## Abstract

**Background/Objectives:** To propose criteria for retreating previously embolized PAVMs and determining the effectiveness of the criteria to prevent paradoxical embolization. **Methods:** A retrospective review of patients with PAVMs treated at a single HHT center of excellence between 1 January 2013, and 10 September 2023, was performed. Patients with PAVM recurrence were either retreated or observed based on the following criteria for PAVM reintervention: 1. Embolic device(s) not creating a sufficiently dense matrix, such that a channel through them may be >/ 2 mm; 2. Accessory feeding artery or pulmonary collateral >/ 2 mm; 3. Hemoptysis in a patient with no other explanation. **Results:** A total of 438 PAVMs were treated in 151 patients, including 106 patients with definite, 14 possible, and 31 doubtful HHT. Post-embolization PAVM recurrence occurred in 36 patients (36/151, 23.8%), including 15 patients (15/151, 9.9%) with 22 PAVMs (22/438, 5.0%) meeting criteria for reintervention. A total of 21 patients (21/151, 13.9%) with recurrence did not meet reintervention criteria and were therefore observed. Pre-treatment paradoxical embolization occurred in 36 patients (36/151) for a lifetime prevalence rate of 23.7%. Post-treatment paradoxical embolization did not occur in any patients following PAVM embolization (0/151). There was one case of iatrogenic paradoxical embolization in a patient being treated for systemic collateral reperfusion and hemoptysis. However, this was not included given that it was not a spontaneous event. **Conclusions:** Utilizing modern embolization techniques and devices, the proposed reintervention criteria, and screening intervals, paradoxical embolizations can be effectively prevented in patients with PAVMs.

## 1. Introduction

Over the past two decades, treatment of pulmonary arteriovenous malformations (PAVMs) has continued to improve. In brief, PAVMs are abnormal connections between pulmonary arteries and pulmonary veins, bypassing the normal pulmonary capillary bed [1,2]. Approximately 80–90% of PAVMs occur in patients with hereditary hemorrhagic telangiectasia (HHT), an autosomal dominant disorder occurring in approximately 1 out of 5000 people [3,4]. In addition to reducing oxygenation, the lack of intervening capillary bed in a PAVM may result in paradoxical embolization. These paradoxical emboli can cause stroke, systemic emboli, or septic emboli [2].

The current standard indication for the treatment of PAVMs is a feeding artery >/ 2 to 3 mm [2,5,6]. Improvements in technique, such as the embolization of the feeding artery <1 cm from the sac and embolic devices such as endovascular plugs has reduced PAVM recurrence rates [5,7]. However, despite use of these techniques and tools, as demonstrated in this study, recurrence of previously treated PAVMs continues to occur. Guidelines for retreatment of recurrent PAVMs is not as well established. The purpose of PAVM embolization is to prevent paradoxical embolization and, in some cases, improve pulmonary oxygenation. The clinical impact of recurrent PAVMs is difficult to evaluate, as they are often found in conjunction with new untreated PAVMs. This topic was explored at the host’s institution in an earlier study from 2006 to 2007 [2]. Furthermore, hypoxia is a multifactorial pathology for which PAVMs are only a part of.

Despite the expansion of HHT treatment centers and improvement in PAVM treatments, much of HHT management is not yet standardized. At the author’s HHT center, post-embolization paradoxical embolization rates have essentially been kept to zero. The purpose of this paper is to therefore report the screening guidelines, intervention techniques, and especially the reintervention criteria utilized to achieve this rate.

## 2. Materials and Methods

### 2.1. Patient Selection and Data Collection

With institutional review board (IRB) approval, a search for patients with PAVMs treated at a single HHT center of excellence between 1 January 2013 and 10 September 2023 was performed. All ages and genders were included. Patients who were embolized only before this time period were not included. The study therefore included lesions that were not treatment naïve as long as they were embolized during the study period. Patients without post-procedural follow-up were excluded. Inclusion criteria is illustrated in the flowchart (Figure 1). The records of these remaining patients were then retrospectively reviewed for demographics, family history, HHT status based on Curacao criteria [4,8], genetic testing, and episodes of paradoxical embolization occurring before and after PAVM embolization. This included episodes of stroke, brain abscess, myocardial infarction, systemic emboli, or septic emboli, for which another etiology was not suspected. Specifically, stroke secondary to brain AVMs were not included as paradoxical embolization. Post-procedural follow-up records and imaging were searched for instances of PAVM recurrence.

### 2.2. Screening

All patients underwent screening for pulmonary AVMs, unless specifically referred for known pulmonary AVMs. Screening was performed with contrast echocardiography and if found to be Grade II (6–20 microbubbles) or higher, a follow-up chest CT with intravenous contrast was performed [9]. Chest CTs were protocolled and read by fellowship-trained thoracic radiologists with the exception of a small number of patients with outside studies. In patients with negative echocardiogram bubble studies, repeat echocardiogram screening was performed every 5 years. In patients with positive bubble echocardiograms studies but with negative chest CT, repeat chest CT screening was performed every 5 years. There were few exceptions to the 5-year screening including young patients with single or few pulmonary AVMs who were screened with echocardiograms. Patients found to have PAVMs with feeding arteries >/ 2–3 mm, were then treated with endovascular embolization.

### 2.3. Follow Up

Post-embolization follow up chest CT with intravenous contrast was performed most commonly after six months (between two months to one year). Chest CTs were obtained prior to the six-month period for patient convenience or when additional surveillance was needed for lung nodules or other thoracic findings. Similarly, the follow-up chest CT was extended up to one year for patient convenience or to coincide with additional thoracic surveillance, so long as the patient had no more than one or two PAVMs. After the initial post-procedural follow-up, if there was no evidence of recurrence or new PAVMs which required treatment, the patients were then resumed to five-year interval chest CT screening.

### 2.4. Endovascular Therapy

All patients were treated by the HHT center director and advising author (J.P.). Embolization technique and devices were determined based on lesion morphology. With simple-type PAVMs (single feeding artery and single outflow vein) embolization of the feeding artery as close to the nidus, but not within the nidus, was performed. The exception was PAVMs with short feeding arteries and unstable catheter position. In these cases, embolization of the nidus was performed. In complex PAVMs (multiple feeding arteries and/or veins), embolization of the common nidus was performed followed by embolization of as many feeding arteries as possible. Vascular plugs were utilized, including Amplatzer Vascular Plugs Type II and IV (Abbott, Chicago, IL, USA) and Microvascular plugs (Medtronic, North Haven, CT, USA), whenever the feeding artery could accommodate these devices. After vascular plug deployment, if complete occlusion was not achieved, additional coils were deployed. In feeding arteries which could not accommodate vascular plugs due to tortuosity, landing zone length, and/or vessel diameter, coils were deployed. Coils included pushable and detachable coils and fibered and non-fibered coils. Pushable fibered coils included Nester (Cook Medical, Bloomington, IN, USA), Tornado (Cook Medical), MReye (Cook Medical), and Hilal coils (Cook Medical). Detachable coils included POD (Penumbra, Inc., Alameda, CA, USA), Azur (Terumo Interventional Systems, Somerset, NJ, USA), Ruby (Penumbra, Inc.), Concerto (Medtronic), Interlock (Boston Scientific Corporation, Malborough, MA, USA), and Embold coils (Boston Scientific Corporation). Right heart and pulmonary pressures were obtained before and after embolization. Technical success was achieved in all cases, as defined by angiographic occlusion of the targeted pulmonary AVM.

Patients with diffuse PAVMs, defined as at least one pulmonary segment with diffuse involvement, remain a challenge for many practices and HHT centers [10]. In these patients, macrofistulas measuring greater than 3 mm were targeted for embolization. This may be divided into multiple treatment sessions if needed. Similarly, in patients with extensive multi-focal disease, PAVMs with >3 mm feeding arteries were embolized. As there is not an exact definition for “extensive” multi-focal disease and in many cases, it is difficult to differentiate between the two, these patients are grouped together as patients for whom endovascular therapy cannot achieve significant improvement in hypoxia and for whom targeting smaller PAVMs with 2–3 mm feeding arteries is not feasible.

### 2.5. Recurrence

PAVM recurrence, also commonly referred to as persistence, was diagnosed on contrast-enhanced chest CTs by fellowship-trained thoracic radiologists. Consistent with prior studies, this was defined as a less than 70% reduction in PAVM sac size or contrast enhancement of the sac on follow-up pulmonary CTA [11]. Recurrences were further divided into PAVMs requiring reintervention based on the below criteria and PAVMs not meeting this criteria. This included previously treated lesions that were now either recanalized (persistent flow through the embolic), filled beyond the coils by accessory feeding arteries, which were either missed on initial embolization or grew, filled beyond coils by pulmonary collaterals, or filled beyond the coils by systemic collaterals.

The following criteria were used by J.P. to determine which patients required reintervention:Embolic device(s) are not creating a sufficiently dense matrix on CT, such that a channel through them may be >/ 2 mm.Accessory feeding artery or pulmonary collateral >/ 2 mm, as measured on CT.Hemoptysis in a patient with no other explanation.

These criteria were confirmed angiographically, with criteria 1 demonstrated angiographically as recanalization (Figure 2a,b). Criteria 2, combines accessory feeding arteries and pulmonary collaterals together as they are generally difficult to differentiate angiographically (Figure 3a,b). Criteria 3 is angiographically shown as perfusion of the PAVM outflow via injection of systemic arteries (Figure 4a,b).

### 2.6. Statistical Analysis

Paradoxical embolization lifetime prevalence and annual incidence were calculated both before and after PAVM embolizations. The lifetime prevalence rates were calculated by dividing the total number of incidences by the number of patients (N events/N patients). The annual incidence was calculated by dividing the total number of events by the number of patient years (N events/N patients × Patient age). A two-tailed t-test was performed using Python (SciPy v1.13.0) to compare paradoxical embolization rates before and after PAVM embolizations. This was further stratified to patients that required reintervention and those that did not.

## 3. Results

During the study period, 438 PAVMs were treated in 151 patients with available follow-up. There were 106 patients with definite HHT, 14 with possible HHT, and 31 with doubtful HHT. This included 97 females (64.2%) and 54 males (35.8%) with an average age of 55 ± 17 years old (18–92). There were 61 patients with ENG mutations, 9 ACVR1 mutations, 2 SMAD4 mutations, and 79 patients without genetic testing. Brain AVMs were present in 10 patients (6.6%). The mean number of lesions treated per patient was 2.9 ± 2.9 (1–17). Two or more lesions were treated in 82 patients (54.3%), and 69 patients had only one lesion treated (45.6%). There were 7 patients with 10 or more PAVMs treated (4.6%).

Post-embolization PAVM recurrence occurred in 36 patients (36/151, 23.8%) and 43 PAVMs (43/438, 9.8%). This included 15 patients with reintervention based on the above criteria (15/151, 9.9%), and 22 PAVMs (22/438, 5.0%). Of the 22 retreated PAVMs, 15 were due to recanalization (15/22, 68.2%), 4 due to systemic collateral reperfusion (4/22, 18.2%), and 3 due to pulmonary collateral reperfusion (3/22, 13.6%). The remaining 21 patients (21/151, 13.9%) with recurrence did not meet criteria for reintervention and were observed. The mean imaging and clinic observational time period for patients without recurrence was 27.0 ± 26.8 months (1–124 months) and 46.5 ± 32.4 months (1–127 months), respectively. Of the 15 patients requiring reintervention, 14 (93.3%) had prior chest CTs showing the persistent occlusion of PAVM(s). The mean imaging and clinic observational time period for patients with recurrence was 39.1 ± 36.9 months (1–108 months) and 65.4 ± 32.2 months (1–110 months), respectively (Table 1—Summary of results).

Pre-treatment paradoxical embolization occurred in 36 patients (36/151) for a lifetime prevalence rate of 23.7%. This occurred over 7236 lifetime years, for an annual incidence of 0.05%. Post-treatment paradoxical embolization did not occur in any patients following PAVM embolization (0/151), including those with a recurrence over 563 lifetime years of post-embolization observation. There was one case of iatrogenic paradoxical embolization in a patient being treated for systemic collateral reperfusion and hemoptysis. However, this was not included given that it was not a spontaneous event. *t*-test analysis demonstrated that these difference in paradoxical embolization rate before and after embolization was meaningful for all patients (T = 12.27, *p* = 0.000), patients without recurrence (T = 11.85, *p* = 0.000), and with recurrence (T = 6.84, *p* = 0.000).

## 4. Discussion

Advancements in PAVM embolization devices and techniques have led to a reduction of PAVM recurrences. This study, to our knowledge, represents the largest cohort evaluating the effectiveness of modern embolization techniques, screening intervals, and reintervention criteria, in preventing paradoxical embolization. It is difficult to determine to what degree the prevention of paradoxical embolization can be attributed to the initial embolization versus the retreatment of recurrent PAVMs. There is some suggestion that the filtration effect of the initial embolization may prevent paradoxical embolization, and in particular, embolization with plugs. However, this has not been reliably shown in the literature yet. In fact, there are some reports that recurrent PAVMs can lead to paradoxical embolization [12]. Current guidelines recommend post-embolization follow-up after 6 months and then 3–5 years thereafter [6]. In this study, the average imaging observational period was 27 months for patients without recurrence and 39 months for patients with recurrence. The longer observational period for the recurrence group was due to the increased imaging in these patients (i.e., a patient without recurrence would have a follow-up only at the five-year mark during the study, whereas a patient with recurrence would have a follow-up at the five-year and then six-year mark). Ultimately, it is difficult to know the ideal follow-up periods for patients, as performing randomized studies may be ethically questionable. Therefore, as a large HHT center, this study offers a combined methodology of modern embolization devices and techniques, screening intervals, and reintervention criteria, which has effectively prevented paradoxical embolization.

Beginning in the early 1990s, embolic coils transitioned from stainless steel to nitinol and other improved metals with greater packing and thrombogenicity. Additionally, there has been an increased use of plugs since the 2000s, which has been showed to significantly reduce recurrence. A good summary of modern PAVM embolization techniques employed at many HHT centers was written by Treotola et al. [5]. Utilizing these modern interventional tools and techniques, as well as the previously described screening intervals and proposed re-intervention criteria, post-treatment paradoxical embolization was effectively prevented in all patients. While pre-treatment paradoxical embolization rates were shown to be significant (life-time prevalence of 23.7% and annual incidence of 0.05%), the prevalence of post-treatment paradoxical embolization was 0. This included no cases of paradoxical embolization in patients with PAVM recurrence, except for one patient with iatrogenic stroke.

The proposed reintervention criteria were based off the greater than 30 years of experiences at a single large HHT center of excellence. The reintervention criteria described in this paper may be used as general guidelines but may not always be generalized to each patient and practice. Additionally, as the population in this study only included adults, it may not be applicable to the pediatric population. The first indication of reintervention requires experience evaluating the embolic devices. A contrast-enhanced chest CT must be windowed in such a way to see the degree to which contrast flows through the embolic device(s). If a column of contrast is measured to be approximately greater than 2–3 mm through, this would suggest a need for reintervention (Figure 2a,b). This may be difficult to determine, and evaluation of the scout images can be helpful to see how well the coil is packed. The second indication for reintervention involves the treatment of accessory pulmonary collaterals (Figure 3a–e). Accessory pulmonary collaterals are a result of adjacent pulmonary vessels which fill the feeding artery or sac beyond the embolic device(s). When this collateral vessel is greater than 2–3 mm, reintervention is indicated. The third criteria involve the treatment of systemic collaterals (Figure 4a–c). This is by far the most complicated issue to address and was the cause of the sole case of iatrogenic stroke. Luckily these systemic collaterals are rare, with a lifetime prevalence of 1.0% [13]. The author’s institutions experience treating these systemic collaterals was previously published. Patients with previously treated PAVMs presenting with hemoptysis are often due to systemic collaterals. In some cases, these lesions can be clearly identified on a chest CTA; however, this is not always the case and therefore a clinical presentation of hemoptysis is enough to indicate reintervention and interrogation of the systemic arteries. Understanding the best way to treat these lesions is an ongoing process.

With paradoxical embolization being effectively controlled by PAVM embolization, attention must increasingly turn to screening programs and identifying at-risk patients from an early age. There are additional factors besides paradoxical embolization that may necessitate PAVM embolization, for example, the treatment of hypoxia. However, as hypoxia is a complicated multifactorial issue, this was not included in this study.

## 5. Conclusions

Utilizing modern embolization techniques and devices, the proposed reintervention criteria, and screening intervals, paradoxical embolizations can be effectively prevented in patients with PAVMs.

## Figures and Tables

**Figure 1 jcm-13-06104-f001:**
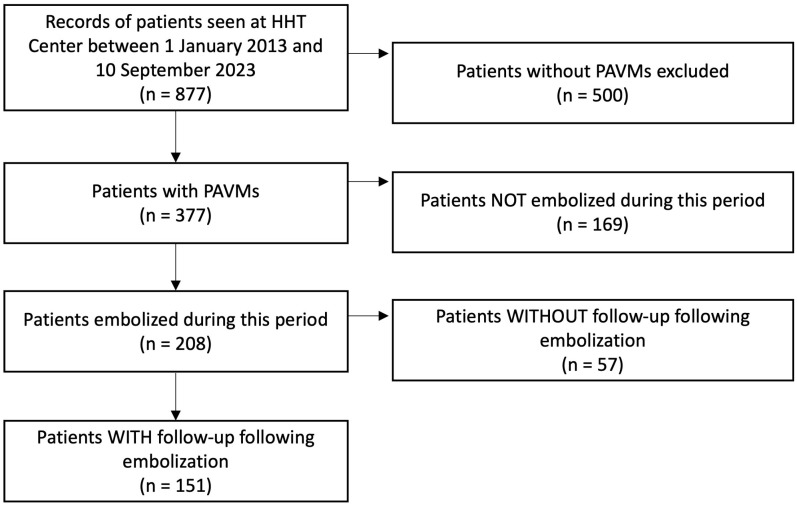
Flowchart illustration inclusion criteria for the study.

**Figure 2 jcm-13-06104-f002:**
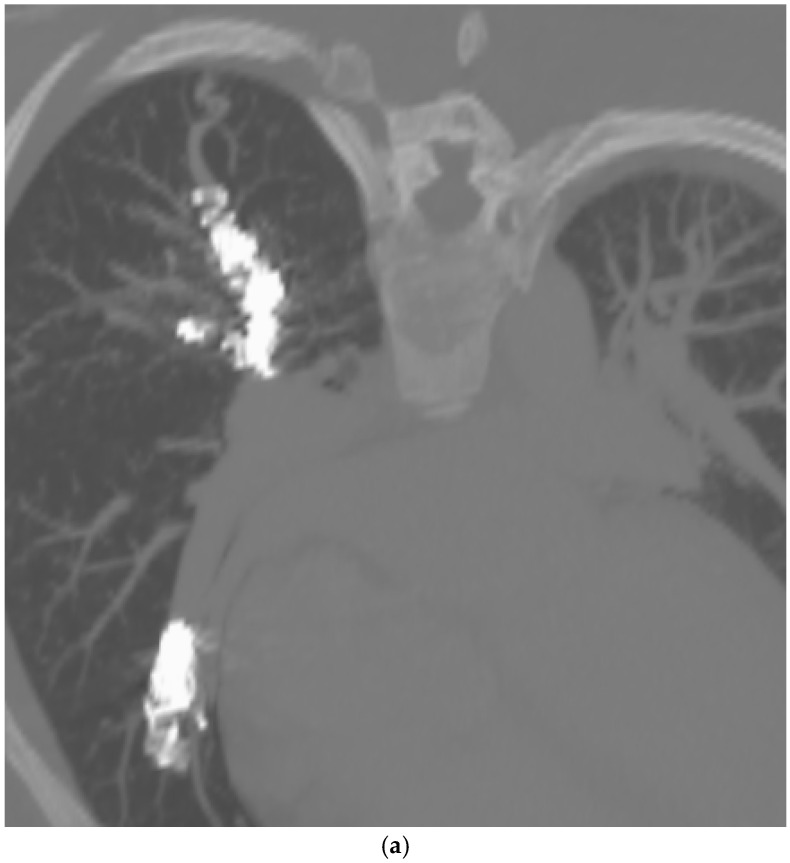
A 32-year-old female with HHT with previous PAVM embolization, presents with worsening extertional dyspnea. (**a**) CT of the chest, oblique anteroposterior view with maximal intensity projection (MIP), demonstrating loosely packed coils resulting in a recanalized RUL PAVM. (**b**) Angiogram of the RUL PAVM confirms recanalization. This was subsequently successfully embolized with a microvascular plug MVP-7Q and two additional coils.

**Figure 3 jcm-13-06104-f003:**
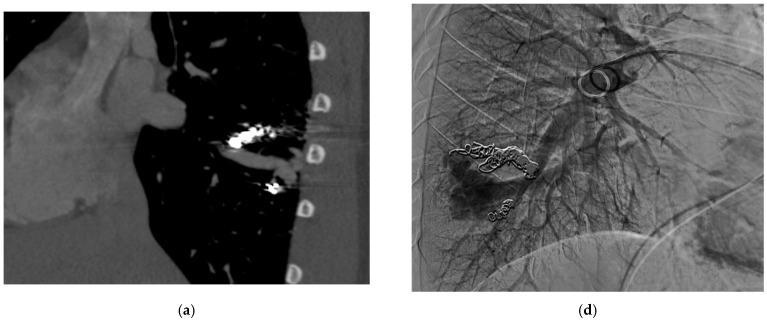
An 18-year-old male with HHT with a diffuse RLL PAVM with at least four feeding arteries previously embolized. Now presents with recurrence identified on surveillance CT. (**a**,**b**) CT of the chest, MIP saggital view, demonstrates multiple collateral pulmonary branches feeding a common outflow channel. (**c**) Angiogram of the right pulmonary artery, early arterial phase, shows multiple collateral arteries feeding the venous outflow, (**d**) seen more clearly on the late arterial phase. (**e**) Repeat angiogram of the right pulmonary artery after embolizing multiple feeding arteries and central venous outflow shows complete occlusion of the RLL PAVM.

**Figure 4 jcm-13-06104-f004:**
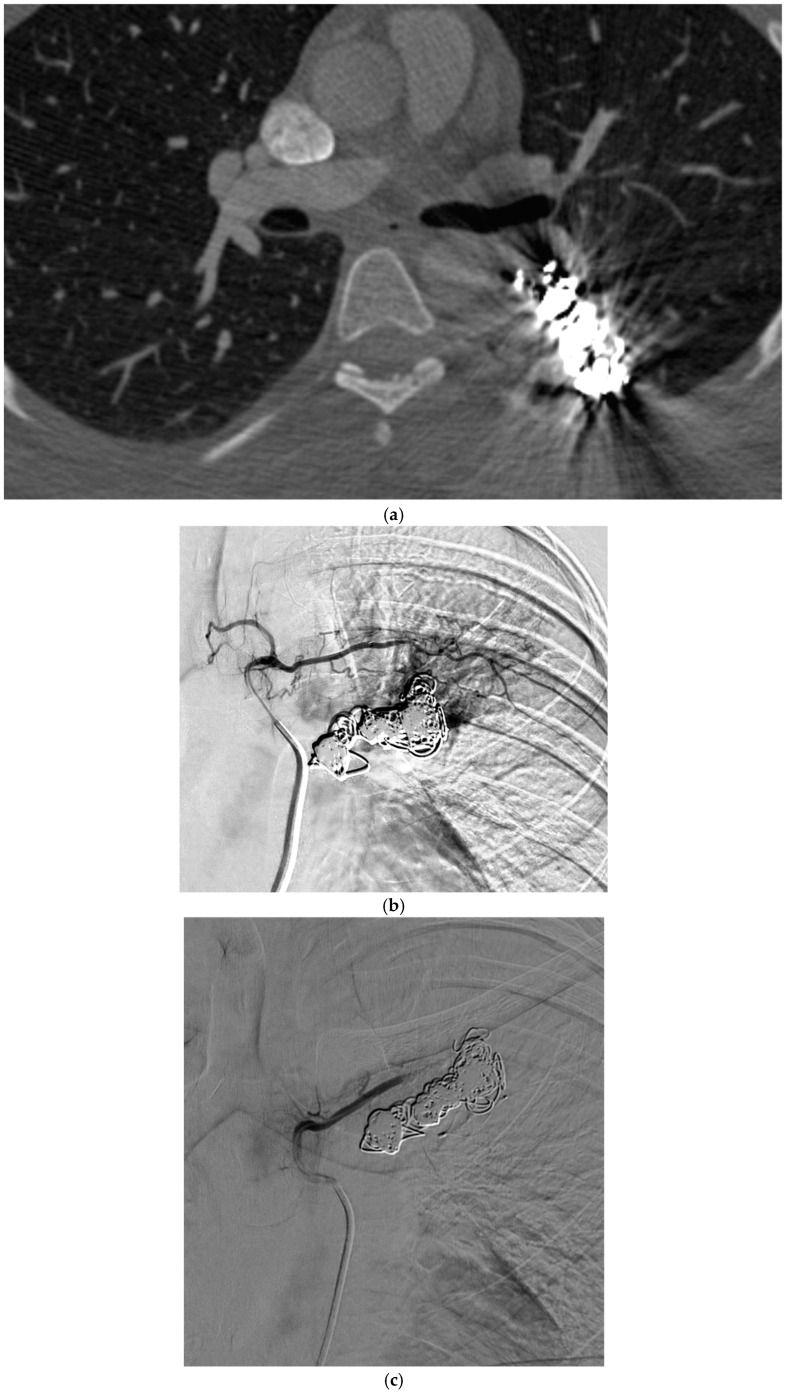
A 26-year-old female with HHT and previously embolized PAVMs, presents with hemoptysis. (**a**) CT of the chest, axial view, demonstrates pulmonary hemorrhage surrounding the embolization coils. (**b**) Pulmonary angiogram demonstrates multiple systemic collaterals arising from a left intercostal artery, filling the venous outflow beyond the previously embolized coils. (**c**) The intercostal artery is successfully occluded with gelatin sponge slurry.

**Table 1 jcm-13-06104-t001:** Summary of recurrence.

	Number of PAVMs	% (N/438 PAVMs)
All recurrence	43	9.8
Recurrence without reintervention	21	4.8
Recurrence with reintervention	22	5.0
Recanalization w/reintervention	15	3.4
Systemic collateral reperfusion w/reintervention	4	0.9
Pulmonary collateral reperfusion w/reintervention	3	0.7

## Data Availability

The original contributions presented in the study are included in the article, further inquiries can be directed to the corresponding author.

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
