# Peer review of "Criteria for PAVM Reintervention"

_jcm, 2024, doi:10.3390/jcm13206104_

Round 1

Reviewer 1 Report

Comments and Suggestions for Authors

This is a well written and interesting and important manuscript, presenting retrospective single center data to support local practices for prevention of paradoxical emboli after PAVM intervention procedures.

The focus was specifically on prevention of paradoxical emboli due to re-perfusion. In our practice, a frequent indication for re-intervention is often due to low oxygen saturation. This can be due to other AVMs that were not treated in the first procedure, or AVMs that grew over time, or due to AVM reperfusion. It would be interesting if this were investigated as well, or at least if the saturation before and after the procedures would be presented. How many additional patients were re-intervened for reasons other than reperfusion? What were the indications? What were the success rates and how is success determined in re-intervention for reasons other than re-perfusion?

Specific points:

Line 66 – please explain “Grade II or higher” or add a relevant reference

Line 69 – “repeat chest CT screening was performed every 5 years” although this is not the focus of this study, it would be interesting to know what the yield of this re-screening time interval was, as it is suggested by some, that this could possibly be extended.

Line 73 – “follow up chest CT with intravenous contrast was performed generally after 6 months.” – would be valuable to have the average/range time interval between the procedure and the imaging.

Lines 80-94 – did you manage to achieve full closure of AVMs in every single procedure? Did you have any cases of diffuse AVMs that could not be fully occluded? Please present this data.

Lines 99-100 – “Recurrences were further divided into PAVMs requiring reintervention based on the below criteria and PAVMs not meeting this criterion” were these criteria implemented in real time? Was this discussed with the patients and documented prior to deciding if intervention is needed?

Lines 129-130 – “The mean imaging and clinic follow-up time-period for patients without recurrence was 27.0 +/- 26.8 months (1-124 months) and 46.5 +/- 32.4 months (1-127 months), respectively” -  did you consider excluding those with less than one year follow up? What clinical meaning does 1 month follow up have for the question of developing paradoxical emboli?  

Line 135 “This occurred over 7,236 lifetime years” – how many lifetime years did you have post treatment?

Discussion and conclusions – please relate to the fact that this is a single centered study and if this can be generalized to other centers (i.e. other radiologist and interventional radiologists) needs to be assessed.

Suggest mentioning that this data relates to adults and it is not clear if can be generalized to the pediatric population.

Figures 1-3  – consider adding images showing the state at end of first intervention

Author Response

This is a well written and interesting and important manuscript, presenting retrospective single center data to support local practices for prevention of paradoxical emboli after PAVM intervention procedures.

The focus was specifically on prevention of paradoxical emboli due to re-perfusion. In our practice, a frequent indication for re-intervention is often due to low oxygen saturation. This can be due to other AVMs that were not treated in the first procedure, or AVMs that grew over time, or due to AVM reperfusion. It would be interesting if this were investigated as well, or at least if the saturation before and after the procedures would be presented. How many additional patients were re-intervened for reasons other than reperfusion? What were the indications? What were the success rates and how is success determined in re-intervention for reasons other than re-perfusion?

Dear Reviewer,

Thank you for taking the time to review this manuscript. Your feedback is sincerely appreciated and we have made a strong effort to include most of your suggestions and critiques. In regards to the paper’s scope, the issue of hypoxia was previously explored by Kaufman

Specific points:

Line 66 – please explain “Grade II or higher” or add a relevant reference.

Thank you for this suggestion, Grade II has been clarified and reference added.

Line 69 – “repeat chest CT screening was performed every 5 years” although this is not the focus of this study, it would be interesting to know what the yield of this re-screening time interval was, as it is suggested by some, that this could possibly be extended.

Agree that this would be interesting. Our follow-up interval is fairly rigid in the 5-year time-frame. There may be some slight variability mostly due to scheduling issues. This could be potentially looked at but it is indeed outside the scope of the paper. We do have some instances of young patients with one or two PAVMs for which we try to spare radiation and follow-up with echocardiogram instead. This has been added to the paper.

Line 73 – “follow up chest CT with intravenous contrast was performed generally after 6 months.” – would be valuable to have the average/range time interval between the procedure and the imaging.

Thank you. We included that the range was 6 months to 1 year.

Lines 80-94 – did you manage to achieve full closure of AVMs in every single procedure? Did you have any cases of diffuse AVMs that could not be fully occluded? Please present this data.

Yes. All targeted PAVMs were embolized successfully. In cases of diffuse or multifocal disease, some smaller PAVMs which we would normally treat were observed instead. This has been added to the paper. Thank you.

Lines 99-100 – “Recurrences were further divided into PAVMs requiring reintervention based on the below criteria and PAVMs not meeting this criterion” were these criteria implemented in real time? Was this discussed with the patients and documented prior to deciding if intervention is needed?

Thank you for this question. This was determined pre-procedurally on CT and documented in pre-op notes. I have clarified that in the paper.

Lines 129-130 – “The mean imaging and clinic follow-up time-period for patients without recurrence was 27.0 +/- 26.8 months (1-124 months) and 46.5 +/- 32.4 months (1-127 months), respectively” -  did you consider excluding those with less than one year follow up? What clinical meaning does 1 month follow up have for the question of developing paradoxical emboli?  

Thank you for this suggestion. We considered this however felt it added bias as it would exclude patients who are immediately post-op and potentially exclude events that may be related to the procedure. Ultimately it is the cumulative sum of time across all patients versus paradoxical embolization rates across all patients that provides the most evidence. 

Line 135 “This occurred over 7,236 lifetime years” – how many lifetime years did you have post treatment?

Thank you for this question, the observation time period was 563 years. We have added that to the paper.

Discussion and conclusions – please relate to the fact that this is a single centered study and if this can be generalized to other centers (i.e. other radiologist and interventional radiologists) needs to be assessed.

Thank you. We have added a line preferencing that reintervention criteria and experiences shared in the paper may not be generalizable to all centers and practices.

Suggest mentioning that this data relates to adults and it is not clear if can be generalized to the pediatric population.

Thank you. We have added a line stating that this may only be applied to the adult population.

Figures 1-3  – consider adding images showing the state at end of first intervention

We can consider adding these figures based on publication preferences, however, initial embolizations are not unique in appearance and may not add much value to the paper.

Reviewer 2 Report

Comments and Suggestions for Authors

General Impression:

This descriptive, retrospective study presents a single institution’s experience with PAVM embolization. The manuscript aims to share the authors' management algorithm, which was associated with an absence of catastrophic post-embolization events. The authors deserve recognition for performing such a large volume of cases. However, there are several limitations that prevent the manuscript from being accepted in its current form:

  1. One of the primary goals of PAVM embolization is the prevention of catastrophic events, such as paradoxical emboli, which have been rarely reported in the existing literature. Some authors suggest that the filtration effect of embolic material in persistent lesions may be sufficient to prevent paradoxical emboli. However, others argue that recanalization and paradoxical emboli, following initial treatment success, may require extended follow-up. In this study, the mean clinical follow-up was approximately five years, with the recanalization group having a longer imaging follow-up compared to the non-recanalization group (27 months vs. 39 months). It would be beneficial for the authors to address this in the discussion section.

  2. Furthermore, anecdotal evidence suggests that paradoxical emboli are more likely to occur in patients with multiple PAVMs, particularly when only a subset of lesions are embolized. Managing bilateral multiple PAVMs is particularly challenging, as some lesions may be too small to treat. If the goal of this manuscript is to provide an algorithm that can be adopted by others, it would be valuable for the authors to elaborate on the management of this subgroup.

Specific comments:

Introduction:

Ok for now

Methods:

Line 55-56: please include a flow chart specifying the exclusion criteria (i.e. xx patients were excluded because of lack of followup).

Line 73: what was the protocol of contrast enhanced CT? CTA PE protocol? Who interpret the CTs? IR or fellowship trained thoracic radiologist?

Line 73: if the purpose of the manuscript is to provide a standard for readers to adopt, authors should avoid using terms such as “generally”. When would author consider ordering a CT earlier than 6 months (vs later than 6mo)?

Line 80: Were all lesions treatment-naive?

Line 83-84: please report the number of simple versus complex lesions.

Line 86-88: please report the number of lesions treated by each device.

Line 92: please report the brand of each coil type.

Line 94: how many patients had multiple PAVMs?

Line 95: please define the difference between “persistence” versus “recanalization”. “Persistence” or “treatment failure” has been commonly defined as less than 70% lesion size reduction. Was all cases of “recanalization” occurred after at least one CT showing “treatment success”?

Line 87-99: please report the number of lesions in each category.

Line 118-119: please provide definition of “possible” and “doubtful” HHT.

Line 126: I assume “recanalization” here refers to persistent flow through embolic?

Line 129-133: please compare follow-up length between recan versus no-recan groups.

Line 141-142: I am not familiar with the statistics used here. But I am not sure if the p value could be 0.00.

Figure 1: please provide a MIP contrast enhanced image in AP view similar to the catheter angiogram. Please also provide DSA. 

Figure 3: what’s the rationale of not using coils, given the risk of spinal ischemia of using gel foam?

Discussion:

To be revised. 

Comments on the Quality of English Language

NA

Author Response

This descriptive, retrospective study presents a single institution’s experience with PAVM embolization. The manuscript aims to share the authors' management algorithm, which was associated with an absence of catastrophic post-embolization events. The authors deserve recognition for performing such a large volume of cases. However, there are several limitations that prevent the manuscript from being accepted in its current form:

  1. One of the primary goals of PAVM embolization is the prevention of catastrophic events, such as paradoxical emboli, which have been rarely reported in the existing literature. Some authors suggest that the filtration effect of embolic material in persistent lesions may be sufficient to prevent paradoxical emboli. However, others argue that recanalization and paradoxical emboli, following initial treatment success, may require extended follow-up. In this study, the mean clinical follow-up was approximately five years, with the recanalization group having a longer imaging follow-up compared to the non-recanalization group (27 months vs. 39 months). It would be beneficial for the authors to address this in the discussion section.

Thank you for this excellent question. We have added more to the discussion on this topic. As you rightly mention we unfortunately do not know the ideal follow up period and whether it is even needed in asymptomatic patients. There are some cases reported of recanalized PAVMs resulting in paradoxical embolization that we cite in the paper. We can therefore only offer the combination of screening, follow up and technique that we use to bring our paradoxical embolization rates to zero. In regards to the mean follow up time. We have clarified that it is more accurate to call this the mean observational time period. Since the study includes patients who have only had their initial post-embolization imaging and clinical evaluation as well as the 5-year follow up and beyond, the mean observational period is therefore less than 5-years. Patients who had reinterventions had additional follow up such that their observational periods within the study were longer. This has been clarified in the discussion.

  1. Furthermore, anecdotal evidence suggests that paradoxical emboli are more likely to occur in patients with multiple PAVMs, particularly when only a subset of lesions are embolized. Managing bilateral multiple PAVMs is particularly challenging, as some lesions may be too small to treat. If the goal of this manuscript is to provide an algorithm that can be adopted by others, it would be valuable for the authors to elaborate on the management of this subgroup.

Thank you for this suggestion, a paragraph has been added to the management of diffuse and extensive multi-focal disease.

Specific comments:

Introduction:

Ok for now

Methods:

Line 55-56: please include a flow chart specifying the exclusion criteria (i.e. xx patients were excluded because of lack of followup).

Line 73: what was the protocol of contrast enhanced CT? CTA PE protocol? Who interpret the CTs? IR or fellowship trained thoracic radiologist?

Images were protocoled and read by thoracic radiologists except for a few patients referred with outside studies. This has been added to the manuscript.  

Line 73: if the purpose of the manuscript is to provide a standard for readers to adopt, authors should avoid using terms such as “generally”. When would author consider ordering a CT earlier than 6 months (vs later than 6mo)?

In some cases patients received chest CTs before the 6 month or up to one-year if they needed surveillance for other thoracic findings or for patient convenience. The range was 2 months to one year. This has been added to the paper.

Line 80: Were all lesions treatment-naive?

Thank you. The lesions were not all treatment naïve. The study included lesions previously embolized as long as they were re-embolized during the study period. This has been clarified in the patient selection paragraph.

Line 83-84: please report the number of simple versus complex lesions. Line 86-88: please report the number of lesions treated by each device. Line 92: please report the brand of each coil type.

Thank you for this suggestion. Our group understands that this is important but it is outside the scope of the paper. In this study, there were 438 PAVMs. Each PAVM was embolized by multiple embolic devices. It would be a very cumbersome process to report each device. Furthermore, as the study reports no cases of paradoxical embolization and has no control or variable group (besides pre- and post-treatment) documenting each embolic device does not really provide value. Instead, we describe the our treatment techniques and strategies. However, based on your suggestion, we have added an extensive list of the brands used by JP to treat PAVMs.

Line 94: how many patients had multiple PAVMs?

Thank you for this question. 54.3% of patients had multiple PAVMs. More details have been added to the results.

Line 95: please define the difference between “persistence” versus “recanalization”. “Persistence” or “treatment failure” has been commonly defined as less than 70% lesion size reduction. Was all cases of “recanalization” occurred after at least one CT showing “treatment success”?

Thank you for this question, we have included the imaging criteria for recurrence or persistence and further defines these terms. 14/15 retreated patients had prior CT showing treatment success, only 1 patient did not. This has been added to the results section. 

Line 87-99: please report the number of lesions in each category. ‘

This has been reported in the results (i.e. Table 1). A paragraph following the criteria has been added to clarify that these criteria match with the types of recurrence.

Line 118-119: please provide definition of “possible” and “doubtful” HHT.

Thank you. This has been defined and referenced in the patient selection and data collection paragraph of the methods section.

Line 126: I assume “recanalization” here refers to persistent flow through embolic?

Yes. A brief clarification has been added in the methods section.

Line 129-133: please compare follow-up length between recan versus no-recan groups.

Thank you. We have provided the follow-up time periods for patients with recurrence and those without. We can additionally provide the time period for patients specifically with recanalization if the reviewers would like this, however I am not sure what the comparison would be to? Other types of recurrences or all patients without recan including those with and without recurrence?

Line 141-142: I am not familiar with the statistics used here. But I am not sure if the p value could be 0.00.

Thank you for this point. We have changed it to p = 0.000 as the p value was 0 to 10 decimal points.

Figure 1: please provide a MIP contrast enhanced image in AP view similar to the catheter angiogram. Please also provide DSA. 

A similar AP view CT MIP has been provided.

Figure 3: what’s the rationale of not using coils, given the risk of spinal ischemia of using gel foam?

Thank you for this question. We have published on this subject before previously. It is because these almost invariably recur and coil embolization will jail you out from future interventions. We had one patient referred to us from an OSH where his bronchial arteries were coil embolized. We could not get past his coils and effectively treat him. He died from massive hemoptysis.

Fish A, Chan SM, Pollak J, Schlachter T: Twenty-Seven-Year Retrospective Review of Hemoptysis from Systemic Collaterals Following Pulmonary Arteriovenous Malformation Embolization. Cardiovasc Intervent Radiol 2023;46:670-674.

Discussion:

To be revised.